# Innovative Drugs First Implemented in Type 2 Diabetes Mellitus and Obesity and Their Effects on Metabolic Dysfunction-Associated Steatohepatitis (MASH)-Related Fibrosis and Cirrhosis

**DOI:** 10.3390/jcm14041042

**Published:** 2025-02-07

**Authors:** Georgiana-Diana Cazac-Panaite, Cristina-Mihaela Lăcătușu, Elena-Daniela Grigorescu, Adina-Bianca Foșălău, Alina Onofriescu, Bogdan-Mircea Mihai

**Affiliations:** 1Unit of Diabetes, Nutrition, and Metabolic Diseases, Faculty of Medicine, “Grigore T. Popa” University of Medicine and Pharmacy, 700115 Iași, Romania; georgiana-diana_cazac@umfiasi.ro (G.-D.C.-P.); elena-daniela-gh-grigorescu@umfiasi.ro (E.-D.G.); giba.adina-bianca@d.umfiasi.ro (A.-B.F.); alina.onofriescu@umfiasi.ro (A.O.); bogdan.mihai@umfiasi.ro (B.-M.M.); 2Clinical Center of Diabetes, Nutrition and Metabolic Diseases, “Sf. Spiridon” County Clinical Emergency Hospital, 700111 Iași, Romania

**Keywords:** MASLD, MASH, liver fibrosis, hepatic cirrhosis, antidiabetic drugs, anti-obesity drugs, GLP-1 receptor agonists, polyagonists, FGF analogs, PPAR agonists

## Abstract

Metabolic dysfunction-associated steatotic liver disease (MASLD), a progressive liver disease frequently associated with metabolic disorders such as type 2 diabetes mellitus (T2DM) and obesity, has the potential to progress symptomatically to liver cirrhosis and, in some cases, hepatocellular carcinoma. Hence, an urgent need arises to identify and approve new therapeutic options to improve patient outcomes. Research efforts have focused on either developing dedicated molecules or repurposing drugs already approved for other conditions, such as metabolic diseases. Among the latter, antidiabetic and anti-obesity agents have received the most extensive attention, with pivotal trial results anticipated shortly. However, the primary focus underlying successful regulatory approvals is demonstrating a substantial efficacy in improving liver fibrosis and preventing or ameliorating cirrhosis, the key advanced outcomes within MASLD progression. Besides liver steatosis, the ideal therapeutic candidate should reduce inflammation and fibrosis effectively. Although some agents have shown promise in lowering MASLD-related parameters, evidence of their impact on fibrosis and cirrhosis remains limited. This review aims to evaluate whether antidiabetic and anti-obesity drugs can be safely and effectively used in MASLD-related advanced fibrosis or cirrhosis in patients with T2DM. Our paper discusses the molecules closest to regulatory approval and the expectation that they can address the unmet needs of this increasingly prevalent disease.

## 1. Introduction

Diabetes mellitus (DM) is common in patients with hepatic cirrhosis. Their coexistence impacts more than one-third of cirrhotic patients, raises the risk of hepatic complications, and increases the risk of hospitalizations and fatalities triggered by decompensation events [1,2,3]. Type 2 DM (T2DM) is also considered an independent risk factor for the development of hepatocellular carcinoma (HCC) [4]. Insulin resistance and prolonged exposure to hyperglycemia can trigger fibrogenic cascades. Subsequently, multiple tissue injuries predominantly based on extracellular matrix accumulation increase the risk of fibrotic lesions in several organs, including the liver [5]. Metabolic dysfunction-associated steatotic liver disease (MASLD), formerly known as non-alcoholic fatty liver disease (NAFLD), is characterized by the presence of steatotic liver disease (SLD) accompanied by one or more cardiometabolic risk factors in the absence of harmful alcohol consumption [4,6]. MASLD includes a wide spectrum of liver conditions, ranging from uncomplicated hepatic steatosis to metabolic-associated steatohepatitis (MASH, previously non-alcoholic fatty liver disease—NASH) with minimal or mild fibrosis, advanced liver fibrosis, liver cirrhosis, and hepatocellular carcinoma [4,7]. The diagnosis of metabolic liver disease relies on the identification of specific metabolic criteria and the exclusion of other causes of hepatic steatosis. The five cardiometabolic criteria encompass increased body mass index (BMI), fasting serum glucose levels, blood pressure, plasma triglyceride concentrations, low HDL-cholesterol levels, or the presence of therapeutic interventions targeting these parameters [4,8]. Liver cirrhosis is the end stage of chronic liver disease, characterized by advanced fibrosis and significant disruption of the normal liver architecture. Liver cirrhosis can determine multiple complications such as portal hypertension, bleeding from esophageal varices, ascites, spontaneous bacterial peritonitis, hepatic encephalopathy, hepatorenal syndrome, and a significant risk for hepatocellular carcinoma (HCC) [9]. Liver fibrosis foretells mortality in MASLD, with or without MASH. Depending on the fibrosis grade, more severe fibrosis portends higher mortality [10]. In turn, a higher age (particularly >50 years), insulin resistance, and the accumulation of multiple cardiometabolic risk factors significantly elevate the risk of developing MASH, severe fibrosis, or cirrhosis. Among these risk factors, obesity and T2DM are associated with the highest risk of both liver-related and all-cause mortality [4,11]. In individuals with compensated cirrhosis, the baseline excess weight (either overweight or obesity) correlates with a higher risk of clinical decompensation, independently of the liver function, portal pressure, or the underlying cause of liver disease [4,12].

A prospective study that assessed 501 patients with T2DM, all above 50 years old, with magnetic resonance elastography (MRE) or vibration-controlled transient elastography (VCTE) found a 65% prevalence of NAFLD, a 14% prevalence of advanced fibrosis, and a 6% prevalence of cirrhosis. In this study group, obesity and insulin treatment augmented the risk of advanced fibrosis but had no association with cirrhosis [13].

Screening for T2DM is recommended for all patients with MASH-related cirrhosis, either using fasting or random blood glucose levels or an oral glucose tolerance test. Glycated hemoglobin A1c (HbA_1c_) is generally a reliable measure of chronic glycemia in T2DM patients. However, it has been found to sub-optimally perform in people with cirrhosis, particularly in decompensated cases, wherein a normal HbA_1c_ does not necessarily exclude T2DM [14,15]. Fructosamine seems a more accurate marker for monitoring glycemic control in these patients, although validated laboratory cut-offs for fructosamine in cirrhotic patients are currently lacking [16,17].

The management of diabetes in patients with hepatic cirrhosis has been intensively researched in the past years, as diagnosis, monitoring, and glycemic control can all be challenging [18]. The benefit of intensive glucose control among patients with cirrhosis remains a topic of debate. Moreover, there is insufficient evidence regarding the optimal antidiabetic strategy, particularly for patients who cannot tolerate metformin or require more aggressive treatment [19]. Consequently, there is a critical need to define the safety and effectiveness of second-line antidiabetic therapies in this high-risk population [15,20]. Limited clinical evidence is also available regarding the safety and efficacy of antidiabetic medications in individuals with cirrhosis, with insulin therapy remaining the safest option for such patients [20,21,22].

Aside from metformin, many conventional antihyperglycemic drugs, such as sulfonylureas, thiazolidinediones (TZDs), and insulin, tend to increase weight. As a result, they may worsen adiposity and potentially aggravate portal hypertension in patients with MASH-related cirrhosis [15,23,24,25]. By contrast, glucagon-like peptide-1 receptor agonists (GLP-1RAs), dipeptidyl peptidase-4 (DPP4) inhibitors, and sodium–glucose cotransporter-2 (SGLT-2) inhibitors either induce weight loss or at least have a neutral effect on weight [8,26]. Therefore, they represent effective alternatives for managing T2DM in MASH. Long-term safety and efficacy data for these medications in cirrhotic patients are still limited, with ongoing randomized controlled trials (RCTs) aiming to address this gap. Nevertheless, for patients with poorly controlled T2DM and decompensated cirrhosis due to MASH, insulin currently remains the primary treatment option [17]. Following local regulatory approvals and label indications, adults with non-cirrhotic MASH and advanced liver fibrosis (stage ≥ 2) should be considered for treatment with resmetirom [27,28]. This agent has shown histological efficacy in addressing steatohepatitis and fibrosis, together with a favorable safety and tolerability profile [4,29]. Introducing effective treatments for MASLD and MASH has proven considerably challenging, since deceleration of the disease progression toward advanced stages is a reasonable requirement [4]. Ongoing trials in MASLD focus on the histological effectiveness of drugs on steatohepatitis, advanced fibrosis, and cirrhosis.

This review aims to collect evidence evaluating whether antidiabetic and anti-obesity drugs can be safely and effectively used in MASLD-related advanced fibrosis or cirrhosis in patients with T2DM. Most content will refer to recently approved or innovative antidiabetic and anti-obesity medications still in the pipeline and their use in advanced stages of liver disease, such as fibrosis and cirrhosis in MASLD. Our paper is the first to concentrate on the effects of these new medications and their perspective on MASH-related fibrosis and cirrhosis, not just hepatic steatosis or steatohepatitis, providing an update based on the latest trials in people with obesity and T2DM.

## 2. Widely Used Antidiabetic Drugs and Their Effects in Patients with T2DM and Cirrhosis

Inadequate glycemic and weight control accelerates the advancement of cirrhosis, and the need to address the various cardiometabolic factors leading to MASLD requires a comprehensive therapeutic strategy, where pharmacological options must display a wide array of positive effects [30].

Current data indicate that metformin, SGLT-2 inhibitors, and GLP-1RAs are promising treatment options for patients with T2DM and compensated liver cirrhosis [31]. These agents provide adequate glycemic control with a low risk of hypoglycemia. Additionally, they offer benefits such as reduced body weight, decreased cardiorenal risk, or even improvement of MASLD [32,33]. In contrast, sulfonylureas are associated with hypoglycemic events, weight gain, and even an elevated risk of HCC, making them unsuitable for these patients. In some studies, DPP-4 inhibitors have been linked to an increased risk of decompensation and variceal bleeding. Therefore, alternative treatments should be prioritized until further evidence becomes available [4,22,34,35]. Although TZDs, such as pioglitazone, have shown effectiveness in patients with T2DM and MASLD with substantial fibrosis (>F2), their use in patients with T2DM is limited by concerns over side effects like edema and osteoporosis [23,36]. Azemiglitazone (MSDC-0602K), another PPAR-γ agonist targeting the mitochondrial pyruvate carrier, seems to be associated with minimal side effects. A phase 2 trial using azemiglitazone in varying doses for MASH patients with F1–F3 fibrosis failed to achieve the primary endpoint of reducing the progression of NAFLD activity score (NAS) without fibrosis. Still, azemiglitazone demonstrated notable metabolic effects [37]. Insulin, which is not hepatotoxic and highly effective in lowering glucose levels, can be safely combined with other medications and is still widely used. However, the risks of hypoglycemia, weight gain, and fluid retention associated with insulin use should be carefully considered [19,21,38]. A phase 2 trial (NCT06135584), which started in China at the end of 2023, aims to evaluate the safety, tolerability, and effectiveness of different drugs in treating MAFLD-related cirrhosis (F3–F4), specifically focusing on their impact on liver function, fibrosis, and overall metabolic health. Participants are randomly assigned to receive pioglitazone-metformin fixed-dose combinations vs. other standard care drugs used alone (pioglitazone, metformin, or GLP-1RAs) and placebo, with close monitoring of liver fibrosis progression and metabolic parameters. The study’s results will hopefully improve therapeutic strategies for cirrhosis linked to metabolic syndrome [39].

SGLT-2 inhibitors may offer a biologically plausible benefit via their mechanism of inhibiting renal sodium reabsorption, which reduces salt and water retention and limits the activation of the renin–angiotensin–aldosterone system [40]. This mechanism could optimize fluid balance and address maladaptive neurohormonal signaling and proinflammatory responses contributing to hepatic decompensation. However, as one of the most recently approved antidiabetic drugs, the safety and efficacy of SGLT-2 inhibitors in cirrhotic patients have not yet been studied and are thus unclear [36]. Among the relatively limited number of SGLT-2 inhibitor studies, the secondary analyses concerning their use in liver cirrhosis are underpowered and should be interpreted cautiously. Thus, large-scale, head-to-head studies are needed to directly compare SGLT-2 inhibitors with other antidiabetic drug classes and evaluate each individual drug in this class.

The drug class most investigated on the grounds of histologic improvement of MASLD is represented by GLP-1RAs. GLP-1 is a hormone the small intestine releases in response to food intake, leading to lower glycemic levels by activating insulin secretion and inhibiting glucagon secretion, and also to delayed gastric emptying and appetite suppression [41]. GLP-1RAs were initially licensed for the treatment of T2DM after evidence of strong abilities to lower HbA_1c_ by controlling the blood glucose and of smaller collateral benefits on lipid profile and blood pressure was obtained. Soon after, GLP-1RAs were approved for the treatment of obesity without diabetes, as they can facilitate a significant degree of weight loss [42,43,44,45]. Representatives of the GLP-1RA class that are currently approved for glycemic control in T2DM include short- and long-acting exenatide, liraglutide, dulaglutide, lixisenatide, subcutaneous and oral semaglutide, efpeglenatide, and beinaglutide [46,47,48]. Seemingly interfering with the link between insulin resistance, adipose tissue, and inflammation, GLP-1RAs have now proven to have the beneficial results of improved liver enzymes and hepatic steatosis, reduced inflammatory markers, and oxidative stress in MASLD and MASH subjects [41,49]. Clinical and preclinical studies have shown an extensive array of promising results in various MASLD patients, with improvements in hepatic steatosis and fibrosis, reduced de novo lipogenesis, and MASH resolution seen vs. placebo. However, data from patients with established liver cirrhosis remain limited, as this group was previously excluded from trials [50]. GLP-1RAs’ use was associated with a reduced risk of mortality and progression to cirrhosis in patients with MASLD and diabetes. However, this protective effect was not observed in those with pre-existing cirrhosis, emphasizing the importance of initiating treatment earlier in the disease progression [51]. The multifactorial effects of GLP-1RAs on MASLD have been previously systematized in Cazac et al., 2023 [52]. Improvement of the endothelial function and reduced vascular inflammation and lipid accumulation can most probably prove beneficial in patients with hepatic cirrhosis. Some other interesting findings can be found in in vivo studies. For example, exenatide seems to show benefits in studies on rats with liver cirrhosis by promoting the protective effects of HOTAIR (long-acting non-coding RNAs lncRNAs HOX Transcript Antisense) on cardiac function in cirrhotic cardiomyopathy [53].

Several clinical studies have focused on the effectiveness of GLP-1RAs compared to other antidiabetic medications in reducing decompensation events among patients with T2DM and cirrhosis. Simon and collaborators were the first authors to retrospectively compare the effects of GLP-1RAs and DPP4 inhibitors, sulfonylureas, or SGLT-2 inhibitors on the risk of hepatic decompensation events (the composite of hospitalizations due to ascites, spontaneous bacterial peritonitis, hepatorenal syndrome, bleeding esophageal varices, or hepatic encephalopathy) in patients with T2DM and cirrhosis [54]. Secondary outcomes were each individual decompensation event. Patients were followed from the initiation of drug treatment until they encountered a first component of the primary outcome, discontinued or changed treatments, lost continuous health coverage, completed the study, or died. A significant limitation was the short follow-up period, with a 132-day median, thus excluding hepatocellular carcinoma as a possible endpoint [54]. The groups receiving GLP-1RAs displayed a significant decrease in the occurrence of hepatic decompensation events compared to those receiving DPP4 inhibitors (1431 matched pairs; HR, 0.68; 95% CI, 0.53–0.88) and sulfonylureas (1246 matched pairs; HR, 0.64; 95% CI, 0.48–0.84). This pattern persisted when individual decompensation events were considered. Paired cohorts that received either GLP-1RAs or SGLT-2 inhibitors showed consistently similar rates of decompensation events (845 matched pairs; HR, 0.89; 95% CI, 0.62–1.28), which persisted in the secondary analysis of individual events. The results remained consistent within the subgroups of patients with and without prior episodes of decompensated cirrhosis. As for the etiology of cirrhosis, more than 60% had NAFLD, 7 to 9% had viral hepatitis, approximately 10% had alcoholic liver disease (ALD), and approximately 20% of cases were of unspecified cause [54]. Longer-term studies are needed, including mechanistic analyses and trials focusing on the efficacy and safety of GLP-1RAs in varying stages of cirrhosis. However, in the limited available data, some studies failed to prove beneficial effects, probably due to small sample sizes of subjects. For example, in a small phase 2 clinical trial, semaglutide did not show a significant improvement in fibrosis or in achieving NASH resolution vs. placebo in 71 patients with NASH and compensated cirrhosis [55].

On the other hand, GLP-1RAs bring along the minor yet notable risk of heart rate elevations, which may potentially raise the risk of variceal bleeding in cirrhotic patients [56]. Another concern refers to patients that require upper gastrointestinal endoscopic procedures for the screening of esophageal varices. As a result of the usage of GLP-1RAs and their effect on gastric motility reduction, sedated patients may potentially aspirate gastric contents still retained in the stomach a long time after meals [57]. The potential impact of GLP-1RA use on the lean body mass (skeletal muscle) and the related physical function in patients with cirrhosis may also prove significant, given an already high risk of muscle wasting (sarcopenia) due to factors like elevated protein catabolism, reduced physical activity, and malabsorption. Although reductions in visceral fat could help mitigate this risk, prospective studies are needed to assess this effect, as careful adjustment of GLP-1RA dosages [49] and regular monitoring of muscle mass and function may be necessary [17,58]. Table 1 summarizes the antidiabetic medications considered safe and approved for use in patients with diabetes and cirrhosis, along with their advantages, side effects, and indications for treatment.

## 3. Newer Antidiabetic and Anti-Obesity Drugs and Their Effects on MASH-Related Hepatic Fibrosis and Cirrhosis

Some new innovative synthetic drugs, either already in use or yet under research, have shown good results in obesity and T2DM and might become useful for the treatment of MASLD, MASH, and subsequent hepatic cirrhosis [59,60]. This section of the review aims to describe the design (and sometimes results) of trials recently published or currently in progress that are expected to shed more light on their potential benefits for MASH-related fibrosis and cirrhosis.

### 3.1. Dual Glucose-Dependent Insulinotropic Polypeptide (GIP)/GLP-1 Receptor Co-Agonists

Tirzepatide, a dual GIP and GLP-1 receptor agonist, also known as a “twincretin,” was approved by the Food and Drug Administration (FDA) in May 2022 to treat T2DM and in 2023 for chronic weight management [61,62]. This approval was based on findings from the Study of Tirzepatide in Participants with T2DM Not Controlled with Diet and Exercise Alone (SURPASS) clinical trials program [63,64]. As a once-weekly subcutaneous injection, tirzepatide was able to lower the HbA_1c_ level by as much as 2.07% compared to placebo at the maximum dose of 15 mg per week [65]. Tirzepatide provides a more intense glycemic control effect than currently available single GLP-1RA therapies [64,66]. A substudy of the SURPASS-3 trial (SURPASS-3 MRI) involving 296 participants evaluated changes in liver fat and other fat compartments over the study period, comparing tirzepatide to insulin degludec [67]. At baseline, liver fat content (LFC) was comparable between the tirzepatide (15.67%) and the insulin degludec (16.58%) groups. By week 52, the 10 mg and 15 mg tirzepatide groups exhibited significantly greater mean absolute reductions in LFC compared to the insulin degludec group (−8.09% vs. −3.38%; *p* < 0.0001). In the tirzepatide groups, reductions in LFC were significantly associated with baseline LFC (r = −0.71, *p* ≤ 0.0006), reductions in visceral adipose tissue (VAT) (r = 0.29), decreases in aspartate aminotransferase (AST) levels (r = 0.33), and reductions in body weight (r = 0.34) [67]. The 52-week SYNERGY-NASH trial (NCT04166773) demonstrated that tirzepatide significantly increased the likelihood of resolution of biopsy-proven MASH compared to placebo in a dose-dependent manner and without worsening of moderate or severe fibrosis [68]. Specifically, 44% of participants in the 5 mg tirzepatide group, 56% in the 10 mg tirzepatide group, and 62% in the 15 mg tirzepatide group achieved MASH resolution, compared to just 10% in the placebo group. The proportion of participants achieving an improvement of at least one fibrosis stage without exacerbating MASH was 55% in the tirzepatide 5 mg group, 51% in the tirzepatide 10 mg group, and also 51% in the tirzepatide 15 mg group, compared to only 30% in the placebo group [69]. As with most tirzepatide studies, this trial did not assess the safety or efficacy of tirzepatide in patients with MASH who had progressed to cirrhosis.

FLAMES (Fibrosis Lessens After Metabolic Surgery), a recently started RCT, will randomly assign approximately 120 patients diagnosed with MASH and liver fibrosis or cirrhosis (F1–F4, as determined by baseline liver biopsy) in a 1:1 ratio to either undergo metabolic surgery or receive medical treatment including liraglutide, semaglutide, or tirzepatide. Participants will be followed for two years, after which a second liver biopsy will be conducted to evaluate the improvement of at least one fibrosis stage without MASH worsening [70].

At this point, tirzepatide seems to be the GIP/GLP-1 receptor co-agonist closest to having enough evidence for a potential recommendation in MASLD and even MASH-related fibrosis.

### 3.2. Dual GLP-1/Glucagon Receptors (GLP-1R/GCGR) Co-Agonists

The dual mechanism of GLP-1R/GCGR co-agonists targets multiple pathways: activation of the GLP-1 receptor in the brain suppresses appetite, thus reducing food intake and weight, and in the pancreatic β-cells, it enhances insulin secretion, improving glucose control. Activation of the glucagon receptor in the peripheral tissues increases energy expenditure, thus promoting a negative energy balance and weight loss; in the liver, it reduces lipid accumulation, oxidative stress, and inflammation while decreasing fibrosis, a hallmark of MASH [71,72,73]. This dual-agonist mechanism concomitantly impacts glucose and lipid metabolism, showing advantages over GLP-1 agonism alone.

#### 3.2.1. Survodutide

In prior phase II trials, survodutide (BI 456906), a once-weekly dual GLP-1R/GCGR agonist, has shown significant efficacy in reducing body weight and hyperglycemia in individuals with obesity and/or T2DM [74,75,76,77]. Sanyal et al. examined the 48-week effects of survodutide in patients with biopsy-confirmed MASH and F1–F3 fibrosis but without cirrhosis (NCT04771273). An improvement in MASH without fibrosis progression was observed in 47% of cases in the 2.4 mg group, 62% in the 4.8 mg group, and 43% in the 6.0 mg group, compared to only 14% in the placebo group. Fibrosis improvement by at least one stage was observed in 34%, 36%, 34%, and 22% of participants, respectively [78]. A recent phase I clinical trial (NCT05296733) assessed the 0.3 mg subcutaneous dose of survodutide in both healthy individuals (with or without overweight/obesity) and in patients with MASH cirrhosis classified in A, B, or C Child–Pugh classes. Survodutide seems to be generally well-tolerated in individuals with compensated or decompensated cirrhosis and does not require dose adjustments based on pharmacokinetics. It also shows the potential for improving liver fat content, liver stiffness, liver volume, body weight, and other hepatic and metabolic disease markers, thus classifying it as a promising candidate for further studies in MASH-related cirrhosis [79].

The LIVERAGE (NCT06632444) and LIVERAGE-Cirrhosis ((NCT06632457) studies are currently ongoing global Phase III clinical trials designed to evaluate the efficacy and safety of survodutide in adults with MASH and stages 2 or 3 of liver fibrosis, as well as in individuals with compensated MASH cirrhosis (stage 4). The two primary endpoints in LIVERAGE are represented by the proportion of patients achieving MASH resolution without fibrosis progression and at least 1-point fibrosis improvement without MASH worsening after 52 weeks of treatment. LIVERAGE-Cirrhosis focuses on the time to first occurrence of all-cause mortality or liver-related events [80,81].

#### 3.2.2. Cotadutide

Cotadutide, a once-daily dual GLP-1R/GCGR agonist, has been studied for its effects on T2DM, kidney disease, heart failure, and MASH [82,83,84,85]. A systematic review and metanalysis concluded that cotadutide outperformed placebo over time in reducing body weight (mean difference [MD] = 3.31 kg, *p* < 0.00001), HbA_1c_ levels (MD = 0.68%, *p* < 0.00001), and fasting plasma glucose levels (MD = 31.31 mg/dL, *p* < 0.00001) [86]. Preclinical models suggested that cotadutide can potentially reduce inflammation, reverse fibrosis, lower HbA_1c_ levels, support weight loss, and improve insulin resistance. These benefits have been subsequently investigated in clinical studies in patients with T2DM [73]. In a 54-week phase 2b trial, cotadutide was associated with favorable changes in body weight, serum alanine aminotransferase (ALT) levels, and markers of hepatic steatosis and fibrosis (NAFLD Fibrosis Score—NFS and Fibrosis-4 Index—FIB-4) in patients with NASH and T2DM [87]. The PROXYMO-ADVANCE trial evaluated the safety and efficacy of subcutaneous cotadutide (in once-daily 300 mg and 600 mg doses) vs. placebo in 74 subjects with obesity, T2DM, and biopsy-proven non-cirrhotic MASH with F1–F3 fibrosis. After 19 weeks, the 600 mg dose significantly improved the hepatic fat fraction (–5.0%) and reduced ALT (–23.5 U/L) and AST (–16.8 U/L) levels, as well as non-invasive indexes of liver fibrosis and metabolic parameters (a 2.34 kg mean weight loss from baseline) vs. placebo [88]. NCT05517226 was a phase I trial that investigated the pharmacokinetics, safety, and tolerability of cotadutide in cirrhotic subjects with mild, moderate, or severe hepatic impairment (Child–Pugh A, B, C), but with no results available [89]. Recently, the development of cotadutide was discontinued due to strategic pipeline considerations. This discontinuation decision was not influenced by any newly observed safety signals or risk/benefit ratio changes. Instead, the producers shifted their focus to AZD9550 [90], a weekly injectable drug in the same class already being evaluated in the CONTEMPO clinical trial (NCT06151964). The CONTEMPO study will assess AZD9550’s effects on overweight or obese patients with or without T2DM and the modifications of their hepatic fat fraction as measured by Magnetic Resonance Imaging Proton Density Fat Fraction (MRI-PDFF) [91].

#### 3.2.3. Efinopegdutide

Another unimolecular once-weekly dual GLP-1R/GCGR agonist is efinopegdutide (MK-6024), which was able to prove significant reductions in body weight compared to placebo, but no reductions of the HbA_1c_ levels, when investigated in people with obesity and T2DM. However, efinopegdutide was also linked to a relatively high incidence of treatment-emergent adverse events, so these indications were discontinued [92,93]. A 24-week use of 10 mg efinopegdutide in people with T2DM and MASLD showed significant LFC improvements (72.7%), outperforming the 42.3% reduction induced by 1 mg semaglutide, despite similar weight losses and a similar tolerability profile [94]. These findings suggest that efinopegdutide could be an effective treatment option for MASLD. Efinopegdutide is currently under investigation for its potential benefits in patients with cirrhosis. A notable study is the phase 2a clinical trial (NCT06465186) aiming to evaluate the efficacy and safety of efinopegdutide in adults with MASH-induced compensated cirrhosis [95]. This trial is currently recruiting participants and seeking to assess how efinopegdutide influences liver fat, inflammation, and fibrosis in this patient population. Additionally, a Phase 2b study (NCT05877547) is currently examining efinopegdutide in 300 non-diabetic individuals with histologically proven pre-cirrhotic NASH [96]. While this study focuses on patients without established cirrhosis, its findings may provide insights into the drug’s potential benefits in earlier stages of liver disease.

#### 3.2.4. Pemvidutide

Other dual GLP-1R/GCGR agonists are being actively studied to treat NAFLD, with Pemvidutide (ALT-801) emerging as a promising candidate for managing obesity and MASH [97]. Harrison et al. conducted a 12-week trial in 94 MASLD patients, which revealed that 65.0%, 94.4%, and 85.0% of patients treated with subcutaneous once-weekly pemvidutide doses of 1.2 mg, 1.8 mg, and 2.4 mg, respectively, experienced a ≥30% relative reduction in hepatic fat assessed by MRI-PDFF, compared to only 4.2% in the placebo group. Additionally, 40.0% (*p* = 0.001), 72.2% (*p* < 0.0001), and 70.0% (*p* < 0.0001) of patients receiving 1.2 mg, 1.8 mg, and 2.4 mg pemvidutide doses, respectively, achieved a reduction in liver fat exceeding 50% [98,99]. Systematic studies assessing the effect of pemvidutide on liver fibrosis have not yet been initiated.

#### 3.2.5. Mazdutide

Mazdutide (IBI362 or LY3305677), a long-acting synthetic peptide analog of oxyntomodulin acting as a dual agonist of the GLP-1 and glucagon receptors, has shown promising results in managing obesity, diabetes, and conditions related to hepatic fibrosis and steatosis. Support for trials in human subjects originated from encouraging data from preclinical studies in mouse models [100]. A meta-analysis comprising data from 680 subjects showed improvements in key metabolic markers, including reductions in body weight, cholesterol, and triglyceride levels [101]. These results made it a strong candidate for the goals of improving cardiometabolic health in patients with diabetes and obesity. Besides the significant weight loss effect demonstrated by the GLORY-1 study (NCT05607680), mazdutide has been shown to counteract the hyperglycemia induced by GCGR activation by adequately balancing the effects of GLP-1R and GCGR activation, overall reducing HbA1c levels and fasting glucose [102,103]. The randomized Phase III clinical study DREAM-1 (NCT05628311) showed a robust glucose-lowering efficacy of mazdutide vs. placebo after 24 weeks. The DREAMS-2 (NCT05606913) study showed superiority of mazdutide in efficacy and safety compared to dulaglutide in 731 Chinese individuals with T2DM who were unable to achieve adequate glycemic control using metformin alone or in combination with other oral antidiabetic medications; these patients received either 4 mg or 6 mg of mazdutide or 1.5 mg of dulaglutide over 28 weeks, meeting the primary endpoint in HbA_1c_ levels from the first part of the study [104]. These effects particularly benefitted T2DM patients, thus positioning mazdutide as a dual-action drug targeting both weight loss and glycemic control. Among the total of 610 Chinese subjects participating in the study, 69 patients with a more than 5% LFC at baseline (assessed by MRI-PDFF) were reexplored after 48 weeks. The mean relative reduction in LFC was −63.3% from baseline for the 4 mg dose and −73.2% for the 6 mg dose, whereas the placebo group experienced an 8.2% increase. Among participants with a baseline LFC of ≥10%, the 6 mg dose led to an 80.2% decrease in LFC by week 48 [105]. All these results suggest that mazdutide may offer a promising alternative for addressing MASLD and related liver issues such as fibrosis and cirrhosis, although no clinical investigation has yet been launched at this time.

As a momentary conclusion, survodutide, cotadutide, and efinopegdutide dual GLP-1R/GCGR co-agonists display potential benefits in MASH-related fibrosis and cirrhosis, while pemvidutide and mazdutide need more investigations in this area.

### 3.3. Triple Hormone Receptor (GLP-1R/GIPR/GCGR) Agonists

Retatrutide (LY3437943) is the first triple agonist targeting GLP-1, glucagon, and GIP receptors [106,107]. Retatrutide significantly improved glycemic control and substantially reduced body weight in individuals with type 2 diabetes, with a safety profile aligned with that of GLP-1 receptor agonists and combined GIP and GLP-1 receptor agonists [108,109]. In another phase 2 trial, 1 to 12 mg weekly doses of retatrutide led to placebo-adjusted weight losses ranging from 6.6% to 22.1% over 48 weeks. Among users of the highest dose, 48% of participants achieved a weight reduction exceeding 25% of their baseline weight, with 26% losing more than 30% [110]. Sanyal et al. conducted another randomized, double-blind, phase 2 trial comparing retatrutide with placebo in 338 adults with obesity. The study included a MASLD substudy involving 98 participants with at least 10% LFC measured by MRI-PDFF. Preliminary data showed that by week 24, normal liver fat levels (<5%) were achieved by none of the participants in the placebo group, compared to 27% in the 1 mg retatrutide group, 52% in the 4 mg group, 79% in the 8 mg group, and 86% in the 12 mg group [111]. Reductions in LFC were strongly associated with improvements in body weight, abdominal fat, and metabolic markers linked to enhanced insulin sensitivity and lipid metabolism. Retatrutide also improved some MASH biomarkers, such as K-18 and Pro-C3 [111,112]. However, for now, limited studies focus on the use of retatrutide in treating cirrhosis, and more research is needed to explore its efficacy and safety in this context.

### 3.4. Long-Acting Amylin Analogs

Cagrilintide, a novel long-acting acylated amylin analog, has been studied alone and in combination with semaglutide for the treatment of obesity and T2DM. REDEFINE trials (NCT05567796, NCT05394519) will probably prove a more pronounced effect on weight reduction, glycemic management, and cardiovascular risk reduction than the use of either drug individually. In individuals with T2DM, the weekly combination of 4.5 mg cagrilintide and 2.4 mg semaglutide (CagriSema) produced a 15.6% reduction in body weight after 32 weeks vs. either semaglutide or cagrilintide alone [113]. Similarly, in patients with overweight or obesity, this combination achieved a 15.4% weight loss at 20 weeks [114]. Both studies lacked a placebo arm, and participants were instructed not to modify their diets or physical activity. The effect of CagriSema is currently assessed in comparison to semaglutide alone or placebo in patients with MASH and significant fibrosis or cirrhosis (NCT05016882), or with alcohol-related liver disease (NCT06409130), with pending results [115,116]. Another ongoing clinical trial (NCT05564104) assesses how cagrilintide is absorbed, distributed, metabolized, and excreted in the body and evaluates its safety and tolerability in subjects with stable liver impairment categorized under Child–Pugh grade A, B, or C cirrhosis vs. healthy participants [117]. Further results are anticipated in the field of advanced fibrosis and cirrhosis.

### 3.5. Oral, Nonpeptide GLP-1RAs

Orforglipron (LY3502970) and danuglipron are oral nonpeptide GLP-1RAs currently under investigation for treating T2DM and obesity [118,119]. In a phase IIb clinical trial evaluating twice-daily danuglipron, over 50% of patients (compared to approximately 40% in the placebo group) discontinued treatment at the highest dose due to gastrointestinal events. However, significant decreases in HbA1c, fasting plasma glucose (FPG), and body weight were observed over 12 weeks [119,120]. In a systematic review and meta-analysis, these two oral GLP-1RAs demonstrated significant reductions in HbA1c (up to 1.3%) and body weight (up to 9.9%) compared to placebo, with an acceptable safety profile and promising metabolic effects [121]. Oral orforglipron, as a nonpeptide partial GLP-1RA, enhances cyclic AMP signaling more effectively than β-arrestin recruitment, which suggests reduced receptor desensitization compared to full GLP-1RAs. Unlike peptide-based oral semaglutide, orforglipron can be taken without regard to food. In a phase 2 clinical trial for T2DM, orforglipron significantly lowered HbA1c levels by 2.1%, compared to a 1.1% reduction with dulaglutide and 0.43% with placebo [122]. Daily 24 to 45 mg orforglipron doses positively promoted weight loss [123,124]. A 36-week post hoc analysis of patients with obesity showed a significant decrease in MASLD-related liver biomarkers (transaminases, Pro-C3, CK-18, leptin) from baseline, with 12, 24, and 45 mg doses of orforglipron [125]. Still, there are no current investigations of this class in MASLD-related cirrhosis.

### 3.6. Dual Chemokine Receptors 2 and 5 (CCR2/CCR5) Antagonists

A potent immunomodulator, cenicriviroc is an oral once-daily dual inhibitor of the chemokine receptors 2 and 5 (CCR2/CCR5). Cenicriviroc was evaluated in a phase IIa study (ORION) involving patients with obesity, insulin resistance, and suspected NAFLD. The ORION study, aiming to examine the impact of a 24-week cenicriviroc treatment on insulin sensitivity, liver enzymes, and liver imaging, reported improvements in multiple glucose parameters [126]. Meanwhile, the CENTAUR trial, a phase IIb study (NCT02217475) comparing the histological effects of cenicriviroc vs. placebo in patients with MASH and fibrosis, demonstrated a reduction in liver fibrosis progression after two years; however, patients with cirrhosis were excluded [127,128,129]. Nonetheless, the phase 3 AURORA study aiming to treat liver fibrosis in adult MASH patients (NCT03028740) was prematurely terminated due to insufficient efficacy, although cenicriviroc was well tolerated and safe in these patients [130,131]. On the metabolic side, Bober et al. reported beneficial effects of cenicriviroc upon pathogenic mechanisms of painful diabetic neuropathy; the drug was able to reduce mechanical and thermal hypersensitivity in both male and female mice and showed more potent analgesic properties than morphine [132].

### 3.7. Thyroid Hormone Receptor Beta (THR-β) Agonists

Resmetirom (MGL-316) is an orally administered selective agonist of the thyroid hormone receptor (THR)-β, specifically targeting the liver. Its activation of THR-β promotes fat metabolism in the liver and helps reduce lipotoxicity in MASH. Resmetirom is the first FDA-approved therapy for noncirrhotic MASH with moderate to advanced (F2-F3) liver fibrosis. Resmetirom is generally well tolerated, with the most commonly observed side effects upon initiation of therapy limited to mild to moderate diarrhea and nausea [133]. In the 52-week, placebo-controlled MAESTRO-NASH trial (NCT03900429) conducted in patients with NASH and liver fibrosis, of which more than 60% had T2DM, both resmetirom doses (80 mg and 100 mg) successfully met the primary goals of steatohepatitis resolution without further fibrosis progression in 30% of patients, compared with only 10% for placebo [29]. Furthermore, the secondary objective of reducing LDL cholesterol was achieved alongside improvements in several invasive biomarkers. Longer-term data from the MAESTRO-NASH and MAESTRO-NASH-OUTCOMES (NCT05500222) trials, which include patients with compensated cirrhosis and T2DM, will be essential for advancing the drug toward regulatory approval [28,29]. The resmetirom usage should be avoided in patients with decompensated cirrhosis Child–Pugh Class B or C [133]. Notably, resmetirom does not count among drugs already approved or under investigation solely for treating T2DM or obesity. Resmetirom is metabolized by cytochrome P4502C8 (CYP2C8), which may contraindicate its use with other CYP2C8 substrates, among which are some glucose-lowering agents like repaglinide and pioglitazone. Consequently, resmetirom might enhance the potential risk of hypoglycemia occurring in patients with T2DM when these drugs are taken concurrently [134].

### 3.8. Fibroblast Growth Factor (FGF) Analogs

#### 3.8.1. Fibroblast Growth Factor 19 (FGF19) Analogs

The therapeutic effects of FGF19 on obesity and T2DM are due to the activation of the FGFR1/β-Klotho complex, which determines weight loss and lowers plasma glucose levels and insulin resistance [135,136]. Aldafermin is a modified FGF19 analog that inhibits bile acid production and regulates metabolic balance. Although the studies in patients with T2DM are yet in an incipient phase, FGF19 seems to be a promising and well-tolerated therapeutic candidate for NASH, having already collected some evidence pointing towards its potential to reduce LFC, fibrosis-related serum biomarkers, and liver enzyme levels [137,138,139,140]. In the ALPINE-4 phase 2b trial (NCT04210245) involving patients with compensated NASH-related cirrhosis, the higher daily dose of 3 mg aldafermin achieved the primary endpoint of significantly reducing the enhanced liver fibrosis (ELF) score from baseline to week 48 [141,142]. Additionally, improvements were seen in other non-invasive tests (NITs). However, no significant histological improvements were noted, even though the authors claimed that the trial design was insufficiently powered to detect them [141].

#### 3.8.2. Fibroblast Growth Factor 21 (FGF21) Analogs

In several preclinical studies, the analog with the most significant potential, FGF21, seems to improve insulin sensitivity, induce weight loss, and lower blood glucose levels [143]. The long-acting Fc-FGF21 fusion protein efruxifermin [144,145], along with the recombinant FGF21 analogs pegozafermin [134,135] and pegbelfermin (BMS-986036) [136], have demonstrated efficacy in improving metabolic parameters and reducing hepatic steatosis and inflammation levels in clinical trials. These treatments seem well tolerated, with manageable side effects limited to diarrhea and nausea. Efruxifermin (AKR-001, AMG 876) is designed to replicate the biological activity of native FGF21, which regulates various metabolic processes. In the ongoing SYMMETRY trial (NCT05039450), a 96-week phase 2b study of efruxifermin in patients with compensated NASH-related cirrhosis, 22% of participants in the 28 mg group and 24% in the 50 mg group achieved the primary endpoint of fibrosis improvement without worsening NASH, compared to 14% in the placebo group [146]. Even though the results were not statistically significant at this early phase, they were associated with improved liver injury and fibrosis NIT [147]. The final results will rely on a follow-up liver biopsy planned after 96 weeks of therapy. A phase III clinical program has been initiated to further evaluate the efficacy and safety of efruxifermin in treating MASH and related conditions. The program includes three distinct studies. The first, SYNCHRONY-HISTOLOGY (NCT06215716), focuses on assessing the effect of two doses of efruxifermin (28 mg and 50 mg) on the primary objective of achieving a ≥1 stage improvement in fibrosis and MASH resolution in patients with F2 or F3 stages of fibrosis [148], and uses non-invasive biomarkers to evaluate the safety and tolerability of efruxifermin in individuals diagnosed with MASLD or MASH [149]. The third study, SYNCHRONY-OUTCOMES (NCT06528314), investigates efruxifermin’s effect on the progression of liver-related and all-cause outcomes in patients with compensated cirrhosis caused by MASH [150]. Together, these studies aim to provide comprehensive insights into the therapeutic potential efruxifermin might have to control liver disease progression and related outcomes.

Pegozafermin (BIO89-100) has demonstrated significant LFC reductions, as measured by MRI-PDFF, in both the phase Ib/IIa and phase IIb ENLIVEN trials. The phase Ib/IIa trial included 81 patients with MASLD who underwent a 12-week treatment period [151]. The least squares (LS) mean relative reductions in LFC compared to placebo were −46.7% for the 3 mg once-weekly dose, −59.4% for the 9 mg once-weekly dose, −45.9% for the 18 mg once-weekly dose, −70.2% for the 27 mg once-weekly dose, −53.0% for the 18 mg once-every-two-weeks dose, and −60.2% for the 36 mg once-every-two-weeks dose. Similarly, the phase IIb ENLIVEN trial, involving 222 patients with MASH and F2/F3 fibrosis stages, demonstrated comparable efficacy following a 24-week treatment period [152] and achieved its histological endpoints, as assessed by paired liver biopsies [153]. The ongoing ENLIGHTEN clinical trial (NCT06419374) will evaluate the efficacy and safety of pegozafermin in subjects with biopsy-confirmed MASH-related compensated cirrhosis (F4) [154].

FGF analogs have garnered attention for their therapeutic potential in MASH-related fibrosis and cirrhosis due to their ability to target key metabolic and inflammatory pathways in liver disease progression.

### 3.9. Galectin-3 Inhibitors

Galectin-3, a β-galactoside-binding lectin, is upregulated in several metabolic disorders, such as obesity and diabetes, by disrupting the insulin signaling pathways in insulin-responsive organs, therefore playing a role in the pathogenesis of T2DM. As a result, Galectin-3 emerges as a promising candidate for the roles of both a predictive biomarker and a therapeutic target within T2DM management [155,156]. In a subgroup analysis belonging to a previous phase 2b trial (NCT02462967), belapectin (GR-MD-02), a galectin-3 inhibitor, was shown to reduce hepatic venous pressure gradient (HVPG) and the development of esophageal varices in patients with NASH-related cirrhosis and portal hypertension who had no varices at baseline. The effect of belapectin, compared to placebo, on preventing varices after 18 months of treatment has been evaluated in a phase 2b/3 trial (NAVIGATE; NCT04365868) recruiting patients with compensated NASH-related cirrhosis. The results indicated that belapectin might help prevent the development of esophageal varices in patients with NASH-related cirrhosis. These findings form the foundation for the ongoing NAVIGATE phase 3 trial [157,158,159].

### 3.10. Dual Peroxisome Proliferator-Activated Receptor (PPAR) α/γ Agonists and PPAR-Alfa, PPAR-Beta, and PPAR-Gamma (Pan-PPAR) Agonists

Saroglitazar, a dual PPAR α/γ agonist, was first approved in India in 2013 for managing diabetes and dyslipidemia and in 2020 for treating MASH [160,161,162]. Saroglitazar has also received marketing authorization in the United States for treating dyslipidemia and hypertriglyceridemia in patients with diabetes [163]. Until now, it has been shown to effectively reduce transaminase levels and improve fatty liver and liver stiffness (assessed by elastography) in MASLD patients with diabetic dyslipidemia [164,165,166,167]. This drug was confirmed to have hepatic beneficial effects in a placebo-controlled RCT (NCT03061721) named EVIDENCES IV, which involved 106 patients with MAFLD or MASH. A 4 mg saroglitazar magnesium dose significantly reduced LFC (measured via MRI-PDFF), ALT, adiponectin, triglycerides, and insulin resistance [168]. Chaudhuri et al. reported the safety and efficacy of saroglitazar in MASLD patients with or without diabetes, including cases with compensated cirrhosis. In this study, biochemical and elastography parameters improved independently of the weight reduction [169]. Saroglitazar magnesium is currently being evaluated in an ongoing placebo-controlled RCT (NCT05011305) involving 180 participants with MASH and fibrosis. This study primarily aims to assess MASH resolution without worsening fibrosis after 52 weeks of therapy. A secondary objective is to determine if saroglitazar can improve liver fibrosis without exacerbating liver inflammation, steatosis, or ballooning [170,171].

Lanifibranor is an oral pan-PPAR agonist that seems to significantly enhance hepatic and peripheral insulin sensitivity, as demonstrated by reductions in fasting liver glucose production and hepatic insulin resistance (IR) index, as well as by improved insulin-stimulated muscle glucose disposal [172]. Currently, it is the only phase 3 drug to show efficacy in achieving both NASH resolution and fibrosis improvement. Confirmation of these outcomes was based on the histological results of the NATIVE trial (NCT03008070), even though the study was limited to non-cirrhotic NASH patients [173]. The orally administered 100 mg/kg/day lanifibranor dose improved fibrosis and reduced portal hypertension in preclinical models of decompensated cirrhosis [174]. An upcoming phase 3 trial (NATiV3, NCT04849728) is set to randomize approximately 1000 patients with NASH and advanced fibrosis [175,176].

Dual and pan-PPAR agonists currently hold the status of promising solutions for treating MASH-related fibrosis and cirrhosis due to their capacity to target metabolic, inflammatory, and fibrotic pathways. Their ability to simultaneously address multiple pathogenic mechanisms makes them attractive candidates for patients with advanced liver disease. However, more research is needed to confirm their safety and efficacy in cirrhotic populations.

### 3.11. Combination Therapies

Growing evidence suggests that combining drugs with different mechanisms of action may be more effective for treating MASH than using single agents. A phase 2 trial is currently evaluating semaglutide, both alone and combined with a fixed dose of cilofexor, a nonsteroidal farnesoid X receptor (FXR) agonist, and firsocostat, a liver-targeted acetyl-CoA carboxylase (ACC) inhibitor, in patients with compensated MASH-related cirrhosis [177]. In patients with MASH and bridging fibrosis or cirrhosis, 48 weeks of combined treatment with cilofexor and firsocostat have previously reduced the MASLD activity score and improved fibrosis biomarkers, thus pointing towards a possible antifibrotic effect [178].

Table 2 and Table 3 summarize the pharmacological agents presently in development for the treatment of MASLD-related fibrosis (F1–F3) and MASLD-related cirrhosis (F4). All the molecules are sorted in Figure 1 by their most advanced RCT phase that is currently available. Figure 2 summarizes the potential pathways making these innovative drugs useful in MASH-related fibrosis and cirrhosis. Among these beneficial effects, most evidence points towards reducing adiposity, decreasing liver inflammation, altering pathways related to oxidative stress and autophagy, and reducing hepatic stellate cell activation, which leads to lower extracellular matrix deposition and, potentially, fibrosis regression [31].

## 4. Main Challenges and Needs in MASH-Related Cirrhosis Clinical Trials

Managing T2DM in patients with compensated cirrhosis poses challenges due to a heightened risk of hypoglycemia, altered pharmacokinetics, and insufficient evidence regarding the risk–benefit ratio of various drug classes. Conversely, poor glycemic control is known to accelerate liver cirrhosis progression. The frequent coexistence of MASLD and T2DM underscores the importance of addressing excess weight and cardiovascular risk factors as part of a comprehensive, multifactorial treatment approach. Cirrhosis in MASLD is often under-recognized compared to other etiologies, with fewer than one in four individuals with cirrhosis receiving proper HCC surveillance [194]. These shortcomings must be prioritized in future clinical and research initiatives [4].

As MASLD’s natural history may include a high potential for reversibility, the hope of obtaining reversal of more advanced fibrosis stages seems reasonable and gives grounds for the extensive research resources currently invested in this direction. The most solid expression of such positive results would be the process of regaining compensation in patients with MASLD-related cirrhosis. Due to encouraging evidence accumulating in the last years, recompensation is presently considered achievable and represents an ambitious yet relatable therapeutic goal for patients with MASLD-related cirrhosis [195].

The above-discussed therapies have shown benefits in reducing hepatic fat content and alleviating fibrosis and cirrhosis, particularly in patients with diabetes and obesity. However, significant evidence gaps still persist regarding their safety in decompensated cirrhosis, long-term effectiveness, and disease-specific outcomes, including the prevention of hepatocellular carcinoma. Future research should focus on assessing these agents in advanced liver disease, together with investigating cardiovascular effects and evaluating cost-effectiveness to optimize treatment strategies for this high-risk population.

## 5. Conclusions and Future Directions

The clinical importance of our review is rooted in its detailed analysis of the efficacy and safety of innovative agents first studied in patients with metabolic diseases and then in advanced metabolic liver disease, including hepatic cirrhosis. Additionally, another of our goals was to highlight the many existing knowledge gaps that await evidence from RCTs, which need to fully address all aspects of care for individuals with obesity, T2DM, and compensated cirrhosis. Clinical knowledge based on RCT results will ultimately enhance the quality of life and improve long-term outcomes for these patients. Until such evidence becomes available and comprehensive, early intervention is essential to prevent the progression of fibrosis and the emergence of cirrhosis in individuals with MASH. This approach, best supported by medical work within multidisciplinary teams, includes obesity treatment through lifestyle changes and anti-obesity medications, as well as management of T2DM with antidiabetic agents shown to reverse steatohepatitis and halt fibrosis effectively.

Until robust clinical trial data reveal the ideal therapeutic regimen for individuals with obesity, T2DM, and compensated cirrhosis, we should focus on finding the critical therapeutic gaps in addressing this growing public health challenge.

## Figures and Tables

**Figure 1 jcm-14-01042-f001:**
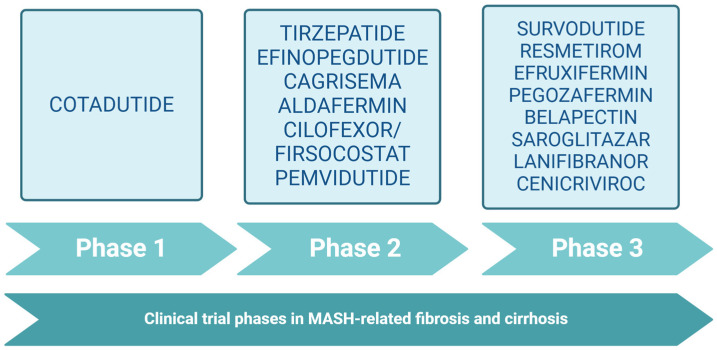
Innovative molecules currently assessed for MASH-related fibrosis and cirrhosis, sorted by their most advanced randomized clinical trial phase.

**Figure 2 jcm-14-01042-f002:**
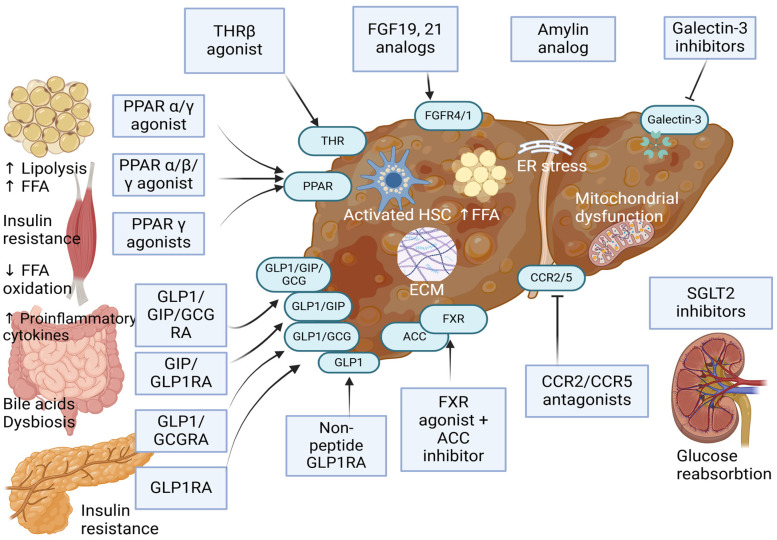
Potential mechanisms of antidiabetic and anti-obesity drugs evaluated in MASH-related fibrosis and cirrhosis. Abbreviations: THR, thyroid hormone receptor; FGF19/21, fibroblast growth factor 19/21; FGFR4/1, fibroblast growth factor receptor 4/1; PPAR, peroxisome proliferator-activated receptor alpha, beta, gamma; GLP-1RA, glucagon-like peptide-1 receptor agonist; GCGR, glucagon receptor agonist; GIP, glucose-dependent insulinotropic polypeptide receptor agonist; ACC, acetyl-CoA carboxylase; FXR, farnesoid X receptor; CCR2/5, C-C chemokine receptor type 2/5; SGLT, sodium–glucose cotransporter 2; ER, endoplasmic reticulum; FFA, free fatty acids; HSC, hepatic stellate cell; ECM, extracellular matrix.

**Table 1 jcm-14-01042-t001:** Antidiabetic medication and considerations in cirrhosis [4,32].

Drug	Advantage	Side Effects	Indication
Biguanides	Good glycemic control Reduced risk of hepatocellular carcinoma in some studies	Risk of lactic acidosis in advanced liver disease	Compensated cirrhosis (Child–Pugh A)
Sulfonylureas		Risk of hypoglycemia Weight gain	Caution in compensated cirrhosis
Thiazolidinediones	Benefits MASH without cirrhosis	Risk of fluid retention, weight gain, and exacerbation of heart failure	Caution in compensated cirrhosis
DPP-4 inhibitors	Weight-neutral No risk of hypoglycemia	Rare cases of pancreatitis	Safe for use in compensated and decompensated cirrhosis
GLP-1 receptor agonists	Glycemic and weight management Improve liver steatosis and reduce inflammation	Gastrointestinal symptoms Potential risk of pancreatitis	Safe in compensated cirrhosis (Child–Pugh A)
SGLT-2 inhibitors	May improve liver steatosis and reduce fibrosis Cardiovascular and renal protective effects	Genito-urinary infections Risk of dehydration and acute kidney injury Diabetic ketoacidosis (rarely)	Compensated cirrhosis (Child–Pugh A and B) with careful monitoring
Alpha-glucosidase inhibitor	No hypoglycemia risk	Gastrointestinal symptoms	Compensated cirrhosis (Child–Pugh A and B)

Abbreviations: MASH, metabolic dysfunction-associated steatohepatitis; DPP-4, dipeptidyl peptidase-4; GLP-1, glucagon-like peptide-1; SGLT-2, sodium–glucose cotransporter-2.

**Table 2 jcm-14-01042-t002:** Ongoing and completed trials investigating the effects of newer molecules in patients with MASH-related fibrosis (F1–F3) (www.clinicaltrial.gov, accessed 20 December 2024).

Intervention	Clinical Trial Identifier, Acronym, Ref.	Trial Phase	Estimated Enrollment	Start Date	Completion Date	Time Frame	Primary Outcomes	Secondary Outcomes
Tirzepatide vs. Placebo	NCT04166773 (SYNERGY-NASH) [68]	Phase 2	196	November 2019	October 2024	52 weeks	Percentage of participants with the absence of NASH with no worsening of fibrosis on liver histology	Percentage of participants with ≥1 point decrease/increase in fibrosis stage with no worsening of NASH on liver histology Mean absolute change from baseline in LFC by MRI-PDFF
Survodutide (BI 456906) vs. Placebo	NCT04771273 [179]	Phase 2	295	April 2021	December 2023	48 weeks	Improvement from baseline in liver histological findings based on liver biopsy in patients with NASH (NAS ≥ 4, Fibrosis F1–F3)	Improvement and percent change in LFC defined as at least 30% relative reduction in LFC by MRI-PDFF Improvement of fibrosis defined as at least one stage decrease in fibrosis stage and NAS by liver biopsy
Survodutide vs. Placebo	NCT06632444 (LIVERAGE™) [80]	Phase 3	1800	October 2024	December 2031	52 weeks	Resolution of MASH without worsening of liver fibrosis on MASH CRN fibrosis score At least a 1-point improvement in fibrosis stage with no worsening of MASH First occurrence of the composite endpoint (progression to cirrhosis, all-cause mortality, liver transplant, hepatic decompensation, worsening of MELD score to ≥15, progression to CSPH	Absolute change from baseline in body weight, HbA1c, ELF score, transaminases, blood pressure, lipids Absolute change from baseline in LSM by TE Improvement of LFC in MRI-PDFF Time to the first occurrence of any of the adjudicated components of the composite endpoint 5P-MACE
Cotadutide 300 μg/600 μg sc once daily vs. placebo	NCT05364931 (PROXYMO-ADV) [180]	Phase 2	54	July 2022	April 2024	28 days	Number of participants with adverse events and abnormal vital signs, or abnormal laboratory assessments	
Efinopegdutide (MK-6024) vs. Semaglutide vs. Placebo	NCT05877547 [96]	Phase 2b	360	June 2023	December 2025	52 weeks	Percentage of participants with NASH resolution without worsening of fibrosis Percentage of participants who experienced an adverse event	Percentage of participants with ≥1 stage improvement in fibrosis without worsening of NASH Change from baseline in body weight
Pemvidutide vs. Placebo	NCT05989711 (IMPACT) [99]	Phase 2	190	July 2023	September 2025	24 weeks	Proportion of subjects achieving NASH resolution with at least a 2-point reduction in NAS without worsening of fibrosis Proportion of subjects achieving at least 1 stage improvement in liver fibrosis without worsening of NASH	Proportion of subjects achieving the composite of both NASH resolution and at least 1 stage improvement of liver Change in LFC by MRI-PDFF Absolute change in MRI-based corrected T1 (cT1) imaging Absolute change in ALT, FAST, ELF
Resmetirom (MGL-3196) vs. Placebo	NCT03900429 (MAESTRO-NASH) [181]	Phase 3	1759	March 2019	January 2028	52 weeks	Resolution of NASH associated with at least a 2-point reduction in NAS without worsening of fibrosis stage Proportion with at least a 1-point improvement in fibrosis stage with no worsening of NAS	Percent change from baseline in directly measured LDL-C
Efruxifermin vs. Placebo	NCT04767529 (HARMONY) [182]	Phase 2b	128	February 2021	March 2024	24 weeks, 96 weeks	Change from baseline in liver fibrosis with no worsening NASH	Resolution of NASH with no worsening of fibrosis Change from baseline in hepatic fat fraction Change from baseline in lipids, HbA1c, HOMA-IR, ELF, Pro-C3, NIS-4, LSM by TE, or body weight
Efruxifermin vs. Placebo	NCT06161571 (SYNCHRONY-REAL-WORLD) [149]	Phase 3	700	November 2023	October 2026	52 weeks	Number of participants with adverse events Number of participants with clinically significant changes in clinical assessments	Change from baseline in ELF, FAST, Pro-C3, LSM by TE, lipoproteins, HbA1c, AST, ALT, GGT, or body weight
Efruxifermin vs. Placebo	NCT06215716 (SYNCHRONY-HISTOLOGY) [148]	Phase 3	1650	December 2023	November 2032	52 weeks, 240 weeks	Resolution of NASH/MASH and a ≥ 1 stage improvement in fibrosis	Resolution of NASH/MASH and no worsening of fibrosis or steatohepatitis Change from baseline in ELF, FAST, Pro-C3, LSM by TE, lipoproteins, HbA1c, AST, ALT, GGT, or body weight
Pegozafermin vs. Placebo	NCT06318169 (ENLIGHTEN-Fibrosis) [183]	Phase 3	1050	March 2024	February 2029	52 weeks	Proportion of participants with at least an improvement of ≥1 stage in fibrosis without worsening of MASH/NASH Proportion of participants with MASH/NASH resolution without worsening of fibrosis	Change from baseline in LFC by MRI-PDFF Change from baseline in ALT
BIO89-100 (Pegozafermin) vs. Placebo	NCT04929483 (ENLIVEN) [184]	Phase 2	222	June 2021	September 2024	24 weeks	Proportion of participants with histological resolution of NASH without worsening of fibrosis Proportion of participants with ≥1 stage decrease in fibrosis stage with no worsening of NASH	Proportion of participants with at least a 2-point improvement in NAS and no worsening of fibrosis Change from baseline in serum lipids, ALT, Pro-C3, HbA1c, or adiponectin Change from baseline in MRI-PDFF
GR-MD-02 (Belapectin) vs. Placebo	NCT02421094 (NASH-FX) [185]	Phase 2	30	September 2015	September 2016	16 weeks	Mean change in liver fibrosis of corrected T1 (cT1) mapping (LiverMultiScan—LMS)	Baseline-adjusted change in LSM with MRE Baseline-adjusted change in LSM by TE
Saroglitazar Magnesium 2 mg vs. Saroglitazar Magnesium 4 mg vs. Placebo	NCT05011305 [171]	Phase 2b	180	August 2021	September 2025	52 weeks	Resolution of NASH with no worsening of fibrosis	Improvement in liver fibrosis with no increase in NAS for ballooning, inflammation, or steatosis Proportion of subjects with ≥1 point improvement in steatosis, ballooning, inflammation, and fibrosis by liver biopsy Change in liver enzyme, ELF, FIB-4, APRI, lipids, HOMA-IR, body weight, HbA1c, IL-6, or CRP
Saroglitazar vs. Vitamin E vs. Combination drug vs. Lifestyle Changes	NCT04193982 [186]	Phase 3	250	January 2021	October 2021	24 weeks	Change in NAFLD fibrosis score	Change in AST, ALT, triglycerides, HbA1c Change in fibrosis and NAS on liver biopsy
Cenicriviroc vs. Placebo	NCT02217475 (CENTAUR) [187]	Phase 2	289	September 2014	June 2017	52, 104 weeks	Number of participants with hepatic histological improvement in NAS by ≥ 2 points with at least 1-point reduction in either lobular inflammation or hepatocellular ballooning and no concurrent worsening of fibrosis	Number of participants with complete resolution of NASH with no concurrent worsening of fibrosis stage or improvement in fibrosis by at least 1 stage (NASH CRN System) and no worsening of steatohepatitis
Tropifexor (LJN452) vs. Cenicriviroc	NCT03517540 (TANDEM) [188]	Phase 2	193	September 2018	October 2020	48 weeks	Number of participants with adverse events	Proportion of participants who have at least a 1-point improvement in fibrosis Proportion of participants with resolution of steatohepatitis
Cenicriviroc vs. Placebo	NCT03028740 (AURORA) [189]	Phase 3	1778	April 2017	March 2021	52 weeks	Percentage of participants with improvement in fibrosis by at least 1 stage and no worsening of steatohepatitis on liver histology Time to the first occurrence of adjudicated events	Percentage of participants with improvement in fibrosis by at least 2 stages and no worsening of steatohepatitis on liver histology
Lanifibranor (IVA337) vs. Placebo	NCT04849728 (NATiV3) [176]	Phase 3	1000	August 2021	September 2026	72 weeks	Resolution of NASH and improvement of fibrosis Safety analyses	

Abbreviations: NASH, non-alcoholic steatohepatitis; LFC, liver fat content; MRI-PDFF, Magnetic Resonance Imaging Proton Density Fat Fraction; NAFLD, non-alcoholic fatty liver disease; NAS, NAFLD Activity Score; MASH, metabolic dysfunction-associated steatohepatitis; CRN, Clinical Research Network; MELD, Model For End-Stage Liver Disease; CSPH, clinically significant portal hypertension; ELF, enhanced liver fibrosis; LSM, liver stiffness measurement; TE, transient elastography, 5P-MACE, 5-point major adverse cardiac event; ALT, Alanine aminotransferase; AST, Aspartate aminotransferase; FAST, FibroScan-AST; LDL-C, low-density lipoprotein cholesterol; GGT, Gamma-Glutamyl Transpeptidase; MRE, magnetic resonance elastography; FIB-4, Fibrosis-4; APRI, AST to platelet ratio index; HOMA-IR, homeostatic model assessment of insulin resistance; IL-6, interleukin-6; CRP, C-reactive protein.

**Table 3 jcm-14-01042-t003:** Ongoing and completed trials investigating the effects of newer molecules in patients with MASH-related cirrhosis (F4) (www.clinicaltrial.gov, accessed 20 December 2024).

Intervention	Clinical Trial Identifier, Acronym, Ref.	Trial Phase	Estimated Enrollment	Start Date	Completion Date	Time Frame	Primary Outcomes	Secondary Outcomes
Tirzepatide vs. Oral Semaglutide	NCT05751720 [190]	Phase 2	30	October 2023	February 2025	52 weeks	Change in liver stiffness in terms of kPa Change in liver fat quantification	Change in liver fat quantification
Roux-en-Y Gastric Bypass/ Sleeve Gastrectomy vs. Tirzepatide/ Semaglutide	NCT06374875 (FLAMES) [70]	Phase 4	120	July 2023	December 2029	2 years	An improvement of at least one stage in the Kleiner fibrosis classification without any progression of MASH in the follow-up liver biopsy	Resolution of MASH and improvement in fibrosis observed in the follow-up liver biopsy
Survodutide vs. Placebo	NCT06632457 (LIVERAGE™—Cirrhosis) [81]	Phase 3	1590	November 2024	June 2029	76 weeks, 4.5 years	Time to first occurrence of the composite clinical endpoint consisting of all-cause mortality, liver transplant, hepatic decompensation events, worsening of MELD score to ≥15, and progression to CSPH	Absolute change in ELF score, body weight, HbA1c, lipids, ALT, AST Change in LSM by TE Time to the first occurrence of progression to CSPH Absolute change in LSM by MRE Time to the first occurrence of any of the composite endpoint 5P-MACE
Cotadutide	NCT05517226 [89]	Phase 1	24	June 2022	February 2023	29 days	Maximum observed plasma (peak) drug concentration Area under plasma concentration time curve from zero to infinity Terminal half-life (t½λz) Apparent total body clearance	Number of participants with adverse events Incidence of ADAs (anti-drug antibodies)
Efinopegdutide (MK-6024) vs. Placebo	NCT06465186 [95]	Phase 2a	80	July 2024	May 2026	28-36 weeks	Change from baseline in LFC Percentage of participants who experienced an adverse event	Change in iron-corrected T1 (cT1) Change in ELF, FIB-4 index, and Pro-C3 Change in LSM by TE Percent change in body weight
NNC0194 0499 vs. Placebo vs. Semaglutide 3 mg/mL	NCT05016882 [115]	Phase 2	672	August 2021	March 2025	52 weeks	Improvement in liver fibrosis and no worsening of NASH	Resolution of NASH and no worsening or improvement of liver fibrosis Change in histology-assessed liver collagen proportionate area Improvement in ballooning and inflammation Change in ALT, AST, hsCRP, ELF, and lipids
Resmetirom vs. Placebo	NCT05500222 (MAESTRO-NASH-OUTCOMES) [191]	Phase 3	700	August 2022	January 2027	36 months	Incidence of composite clinical outcome event (all-cause mortality, liver transplant, ascites, hepatic encephalopathy, gastroesophageal variceal hemorrhage, and increased MELD score from <12 to >/= 15 due to liver disease)	
Aldafermin (NGM282) vs. Placebo	NCT04210245 (ALPINE 4) [142]	Phase 2b	160	March 2020	February 2023	48 weeks	Improvement in ELF score Safety assessed by reported and observed adverse events	
Efruxifermin vs. Placebo	NCT05039450 (SYMMETRY) [192]	Phase 2b	200	July 2021	April 2024	36-96 weeks	Change from baseline in fibrosis with no worsening of steatohepatitis assessed by NASH CRN system	Resolution of NASH with no worsening of fibrosis Change in fibrosis in subjects with no worsening of NASH Change in NIT of fibrosis, body weight, lipoproteins, glycemic control, and HOMA-IR
Efruxifermin vs. Placebo	NCT06528314 (SYNCHRONY-OUTCOMES) [150]	Phase 3	1150	September 2024	October 2029	96 weeks 5 years	First occurrence of disease progression as measured by a composite of protocol-specified clinical events ≥1 stage improvement in fibrosis and no worsening of steatohepatitis	Change from baseline in ELF, Pro-C3, LSM by TE, FAST score, AST, ALT, lipids, HbA1c, HOMA-IR, or body weight
Pegozafermin vs. Placebo	NCT06419374 (ENLIGHTEN-Cirrhosis) [154]	Phase 3	762	May 2024	August 2031	24–60 months	Proportion of participants achieving fibrosis regression	Change from baseline in ELF, ALT, VCTE Proportion of participants who develop CSPH
Belapectin (GR MD-02) vs. Placebo	NCT04365868 (NAVIGATE) [159]	Phase 2b/3	357	June 2020	December 2024	78–156 weeks	Proportion of patients in the belapectin treatment groups who develop new esophageal varices	Cumulative incidence rate of patients who develop esophageal or gastric varices, ascites, spontaneous bacterial peritonitis, or hepatic encephalopathy requiring treatment or hospitalization Cumulative incidence rate of patients who develop liver transplant or MELD score ≥ 15 Change in LSM by TE
Belapectin (GR MD-02) vs. Placebo	NCT02462967 (NASH-CX) [193]	Phase 2	162	June 2015	October 2017	52 weeks	Change in Portal Pressure (HVPG)	Change in the collagen proportional area Change in liver stiffness Percentage of subjects who have at least a one-stage change in liver biopsy histopathological staging of fibrosis Percentage of subjects that develop a clinical complication of cirrhosis
Semaglutide (SEMA) vs. Cilofexor (CILO)/Firsocostat (FIR) vs. PTM SEMA vs. PTM CILO/FIR	NCT04971785 (WAYFIND) [177]	Phase 2	457	August 2021	December 2024	72 weeks	Percentage of participants who achieve ≥ 1-stage improvement in fibrosis according to the NASH CRN classification without worsening of NASH	Percentage of participants with NASH resolution

Abbreviations: kPa, kilopascal; MASH, metabolic dysfunction-associated steatohepatitis; NASH, non-alcoholic steatohepatitis; MELD, Model For End-Stage Liver Disease; ELF, enhanced liver fibrosis; FIB-4, Fibrosis-4; ALT, Alanine aminotransferase; AST, Aspartate aminotransferase; LSM, liver stiffness measurement; TE, transient elastography; CSPH, clinically significant portal hypertension; MRE, magnetic resonance elastography; 5P-MACE, 5-point major adverse cardiac event; LFC, Liver fat content; hsCRP, high-sensitivity C-reactive protein; CRN, Clinical Research Network, NIT, noninvasive tests; HOMA-IR, homeostatic model assessment of insulin resistance; FAST, FibroScan-AST.

## Data Availability

Not applicable.

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
