# Peer review of "Innovative Drugs First Implemented in Type 2 Diabetes Mellitus and Obesity and Their Effects on Metabolic Dysfunction-Associated Steatohepatitis (MASH)-Related Fibrosis and Cirrhosis"

_jcm, 2025, doi:10.3390/jcm14041042_

Round 1

Reviewer 1 Report

Comments and Suggestions for Authors

The review paper is well written, but I have a few comments for improvement.

1.      Abstract: correct part of the first sentence "can progress asymptomatically to liver cirrhosis and, ultimately, hepatocellular carcinoma" as "has the potential to progress symptomatically to liver cirrhosis and, in some cases, hepatocellular carcinoma."

2.      Correct the aim of this review as follows: This review aims to evaluate whether anti-diabetic and anti-obesity drugs can be safely and effectively used in MASLD-related advanced fibrosis or cirrhosis in patients with T2DM. The same goal should be presented in the abstract and the paper's aims.

3.      Some sentences are too long and include multiple ideas, making them difficult to read and understand.  Break complex sentences into shorter, focused statements for clarity. Give a sentence as a conclusion at the end of each part with a subtitle.

4.      Tables: It would be desirable to indicate the name of the drug in the first column of the tables and the registration number of the study in the second column.

5.      Illustrate potential drug mechanisms within a patient with MASLD.

6.      Highlight what is a novelty in this review paper concerning previously published reviews in this field.

7. Specify which areas (e.g., drug safety, long-term effectiveness) require further exploration.

8.      Line 620: "We find ourselves holding limited evidence yet a considerable potential for effective forthcoming treatment change": "We find the critical therapeutic gaps in addressing this growing public health challenge."

"Limited Evidence Yet Considerable Potential": This phrase can appear contradictory, as it simultaneously highlights limited evidence and significant potential.

Reviewer 2 Report

Comments and Suggestions for Authors

1. Originality: The article contains new and important information adequate to justify its publication regarding innovative Drugs First Implemented in Type 2 Diabetes. 2
Mellitus and Obesity and Their Effects on Metabolic
2. Fit to the scientifical literature: The paper demonstrates an adequate understanding of the relevant literature and cite an appropriate list of literature sources.

3. Methodology: The paper's argument is built on an appropriate base of theory and concepts. The research on the paper is well designed and the methods employed are appropriate.

4. Results: The results are presented clearly, concise, and precise.

5. Discussions: The results are analyzed appropriately, and the conclusions are adequate.

5. Implications for research, practice and/or society: The paper clearly identifies the implications for research and practice. These implications are consistent with the findings and conclusions of the paper.

6. Quality of Communication: The paper clearly presents its case, in an appropriate technical language of the field and at the expected knowledge of the journal's readership. The attention has been paid to the clarity of expression and readability, balancing precision and concision.

Reviewer 3 Report

Comments and Suggestions for Authors

The subject of the manuscript is vital in this era where MAFLD?MASH and obesity in addition to DM have extremely high incidence. There are explosion of information available nowadays related to this subject. Many patients start using such a medications, even without doctors orders, pushed by their eagerness to loss weight with or without being diabetics. Many side effects had been recorded due to the huge number of patients using such medications without doctor orders. There should be control and policy for using these medications.

The authors discuss all aspects of the available and in pipe medications, their advantages, the serious side effects and the expected benefits in patients with MAFLD-induced cirrhosis. The design, the introduction and the conclusions are informative.

I need the authors to add a schedule of the only market available and approved medications for use for diabetic patients with cirrhosis, their advantage, the serious side effects, the best indications for treatment.

Introduction:

Line 56: advanced chronic liver disease (ACLD)-----> It denotes active compensated liver disease.
